# *Copaifera langsdorffii* Oleoresin-Loaded Nanostructured Lipid Carrier Emulgel Improves Cutaneous Healing by Anti-Inflammatory and Re-Epithelialization Mechanisms

**DOI:** 10.3390/ijms242115882

**Published:** 2023-11-01

**Authors:** Lucas F. S. Gushiken, Fernando P. Beserra, Maria F. Hussni, Murilo T. Gonzaga, Victor P. Ribeiro, Patrícia F. de Souza, Jacqueline C. L. Campos, Tais N. C. Massaro, Carlos A. Hussni, Regina K. Takahira, Priscyla D. Marcato, Jairo K. Bastos, Cláudia H. Pellizzon

**Affiliations:** 1Hematology and Transfusion Center, University of Campinas—UNICAMP, Campinas 13083-878, SP, Brazil; 2Department of Pharmaceutical Sciences, School of Pharmaceutical Sciences of Ribeirão Preto, University of São Paulo—USP, Ribeirão Preto 14040-903, SP, Braziljkbastos@fcfrp.usp.br (J.K.B.); 3Department of Structural and Functional Biology, Institute of Biosciences of Botucatu, São Paulo State University—UNESP, Botucatu 18618-689, SP, Brazil; 4Department of Veterinary Surgery and Animal Reproduction, School of Veterinary Medicine and Zootechnics, São Paulo State University—UNESP, Botucatu 18618-681, SP, Brazil; 5Department of Veterinary Clinics, School of Veterinary Medicine and Zootechnics, São Paulo State University—UNESP, Botucatu 18618-681, SP, Brazil

**Keywords:** wound healing, skin, nanostructured lipid carrier, emulgel, *Copaifera langsdorffii*, anti-inflammatory, re-epithelialization

## Abstract

The skin is essential to the integrity of the organism. The disruption of this organ promotes a wound, and the organism starts the healing to reconstruct the skin. *Copaifera langsdorffii* is a tree used in folk medicine to treat skin affections, with antioxidant and anti-inflammatory properties. In our study, the oleoresin of the plant was associated with nanostructured lipid carriers, aiming to evaluate the healing potential of this formulation and compare the treatment with reference drugs used in wound healing. Male *Wistar* rats were used to perform the excision wound model, with the macroscopic analysis of wound retraction. Skin samples were used in histological, immunohistochemical, and biochemical analyses. The results showed the wound retraction in the oleoresin-treated group, mediated by α-smooth muscle actin (α-SMA). Biochemical assays revealed the anti-inflammatory mechanism of the oleoresin-treated group, increasing interleukin-10 (IL-10) concentration and decreasing pro-inflammatory cytokines. Histopathological and immunohistochemical results showed the improvement of re-epithelialization and tissue remodeling in the *Copaifera langsdorffii* group, with an increase in laminin-γ2, a decrease in desmoglein-3 and an increase in collagen remodeling. These findings indicate the wound healing potential of nanostructured lipid carriers associated with *Copaifera langsdorffii* oleoresin in skin wounds, which can be helpful as a future alternative treatment for skin wounds.

## 1. Introduction

The skin is the first protective barrier of vertebrates against deleterious agents. When there is a disruption in this barrier, the organism starts the wound-healing process, activating overlapping and dependent mechanisms to reconstruct the skin [1]. Currently, several drugs on the market help with wound healing, avoiding errors, and optimizing the process. One of the drugs used in skin wound healing is a formulation of neomycin and bacitracin used to treat skin lesions to avoid infection [2]; dexpanthenol, which improves the proliferation of keratinocytes and fibroblasts in skin wounds [3]; and collagenase, an enzyme that acts in the wound debridement and remodeling of wounded tissue [4]. However, depending on the location, extent, and type of lesion, the existing treatments may not be efficient [5]. Therefore, several studies have discovered new alternative drugs that accelerate skin wound healing, including medicinal plants and natural products [6].

One of these plants is *Copaifera langsdorffii* Desf. (Leguminosae), popularly known as “copaiba”, whose oleoresin is used in folk medicine to treat skin wounds [7]. In previous studies, our group proved the wound-healing potential of *Copaifera langsdorffii* oleoresin and some mechanisms of action in a 10% cream formulation [8]. However, the cream formulation has low acceptance by patients, and there is poor physicochemical stability with oleoresin [9]. Furthermore, the 10% concentration in topic formulations represents a high amount of active principle and may cause sensitization in chronic treatments. For this reason, researchers have developed new strategies to improve the absorption and effectiveness of drugs through the encapsulation of molecules in nanoparticles, such as nanostructured lipid carriers, which improve the biocompatibility, drug release, and stability of the incorporated drugs [10]. Nanostructured lipid carriers are drug-delivery systems composed of solid lipids with liquid oil, with the synthesis of nanoparticles in the formulation that increase the stability and the capacity of drug release [11]. Therefore, our group tested a new formulation with 1% copaiba oleoresin loaded in a nanostructured lipid carrier to analyze the wound healing potential and mechanisms of action of this formulation in a rat excision wound model.

## 2. Results

### 2.1. Characterization of the Topical Formulation Containing Copaiba Oleoresin-Loaded Nanostructured Lipid Carriers

Nanostructured lipid carrier with 1% *Copaifera langsdorffii* oleoresin (NLC) exhibited a size of 204.06 ± 8.3 nm, a low polydispersity index (0.132 ± 0.028) with narrow particle size profile (Figure 1A), and a negative zeta potential (−17.0 ± 1.7 mV). Furthermore, the nanostructure exhibited spherical morphology as shown in the atomic force microscopy image of Figure 1B. The nanoparticles showed high stability over 60 days (Figure 1A). The stability of these nanoparticles can be attributed to the use of a non-ionic surfactant (Pluronic F68) in conjunction with a negative zeta potential value, which confers steric and electrostatic stability to the nanoparticles [12].

This dispersion was added to an emulgel formulation. The rheological behavior of emulgel containing copaiba oleoresin-loaded nanostructured lipid carrier showed a non-Newtonian character with pseudoplastic behavior (n < 1), which means that with increasing shear rate, the viscosity decreased (Figure 2).

### 2.2. Macroscopic Analysis

The retraction of the wound area was analyzed daily, once a day, for fourteen days, with an increase in the percentage of retraction of all treated groups compared to untreated animals and the best retraction of the group treated with NLC after 3 days. According to the results obtained after 7 days of treatment, the animals treated with Collagenase (Col) and NLC presented better retraction of wounds and no longer had fibrinous exudate. In the macroscopic analysis of 14 days, there was a significative retraction of lesions in the groups treated with Col and NLC compared to untreated animals and the NLC-treated group presented significant results compared to all treatments (Figure 3; Appendix A).

### 2.3. Histopathological Analysis

The histopathological analysis with hematoxylin-eosin (HE) of the total number of cells in the epidermis showed that there were no differences among the wounded groups during days 3, 7 and 14. However, the untreated animals (UT) presented an increase in cells compared to the control (Control: normal skin with no lesion) at 3 and 14 days, indicating that only the untreated group had morphological differences compared to normal skin (Figure 4 and Figure 5). The analysis of HE in the border of the dermis showed an increase in cells in the UT group (3 days), Col and NLC (14 days) compared to the Control (Figure 4 and Figure 6). The histopathological results after 14 days of treatment showed an increase in cells in the UT group compared to the Control in all periods of treatment (Figure 4 and Figure 7). Furthermore, there was a decrease in the thickness of the epidermis in the NLC group compared to the Control at 7 days, and in neomycin + bacitracin (NeBa), dexpanthenol (Dex), Col and NLC treatments compared to the untreated (UT) animals at 14 days (Figure 5 and Figure 8). During the re-epithelialization, there is an enhancement of the thickness of the epidermis. However, the chronic increase in the epidermis thickness and the hyperproliferative state of the keratinocytes. Therefore, the reduction in skin thickness in the groups NeBa, Dex, Col and NLC may indicate a more significant similarity compared to normal skin, unlike the untreated group. The results obtained from the quantification of blood vessels at the border and central region of the lesions did not show any differences among the wounded groups in any period analyzed, with an increase in the number of vessels of UT, NeBa, Dex, Col and NLC groups compared to Control (Figure 6, Figure 7 and Figure 9). 

Histopathological analysis of the collagen amount (greenish color area represented in Figure 10 and Figure 11) in the border of the dermis with Mallory’s trichrome stain presented a reduction in the collagen area in UT, NeBa and Dex groups compared to the Control at 3 days, suggesting that collagenase and NLC-treated groups had a similar amount of collagen compared to normal skin in the border of lesions at 3 days (Figure 10 and Figure 12). The quantification of collagen in the center of wounds showed an increase in collagen content in the NLC-treated group and the Control compared to the UT group (Figure 11 and Figure 12).

### 2.4. Immunohistochemical Analysis

The analysis of α-smooth muscle actin (α-SMA) immunolabeling demonstrated the increase in immunolabeled cells in NLC treatment on days 3 and 7 compared to the untreated (UT) group (Figure 13 and Figure 14). The Control (unwounded skin) expressed low levels of α-SMA. The immunohistochemical analysis of desmoglein-3 showed decreased immunolabeling in the keratinocytes of the NLC treatment during 3 and 7 days compared to the UT group, and the Col group had low immunolabeling compared to the untreated group on day 7 (Figure 13 and Figure 15). After 14 days, the untreated group presented an increased immunolabeled area compared to the Control, suggesting that after two weeks, NeBa, Dex, Col and NLC groups expressed the same normal levels of desmoglein-3 compared to normal skin. According to laminin-γ2 immunolabeling, after 3 and 7 days, it was possible to observe increased immunolabeling with the Col and NLC treatments compared to Control (Figure 13 and Figure 16). After 14 days, it is possible to observe an increase in immunolabeling of laminin-γ2 in the NLC-treated group compared to UT, NeBa and Col groups, and all wounded groups showed increased values than Control (Figure 13 and Figure 16). The data of the cell proliferation marker (Ki-67) in the epidermis, border and center of the dermis did not demonstrate any difference among wounded groups. However, the number of immunolabeled cells from UT, Col and NLC groups was higher than the Control group (Figure 17, Figure 18, Figure 19 and Figure 20).

### 2.5. ELISA

The analysis of interferon-γ (IFN-γ) showed a decrease in the level of this pro-inflammatory cytokine in the NeBa, Dex, Col and NLC after 7 and 14 days, compared to the untreated group. The results of interleukin-1β (IL-1β) and interleukin-6 (IL-6) demonstrated their reduction in animals treated with NeBa, Dex and NLC after 3, 7 and 14 days, and Col-treated group after 7 and 14 days. There were no statistical differences in the concentration of interleukin-10 (IL-10) among wounded groups after 3 days. However, this anti-inflammatory cytokine was higher in NeBa, Col and NLC compared to the Control. After 7 days, all groups presented similar concentrations of IL-10. After 14 days of treatment, the levels of IL-10 were increased in NeBa, Dex, Col and NLC treatments compared to the untreated group. NeBa, Dex, Col and NLC showed reduced levels of tumor necrosis factor-α (TNF-α) compared to the untreated group after 14 days of treatment (Figure 21). The UT group presented higher concentrations of this pro-inflammatory cytokine compared to the Control in all periods analyzed, suggesting an increase in inflammation in the area.

### 2.6. Toxicological Analysis

The results of the systemic toxicity of liver enzymes aspartate aminotransferase (AST), alanine aminotransferase (ALT) and γ-glutamyl transferase (γ-GT), and kidney proteins (creatinine and urea) did not show differences among any treatments and normal parameters (Control), demonstrating the absence of systemic effects of NLC and its safety for further investigations (Table 1).

## 3. Discussion

In previous studies, our group demonstrated the healing potential of copaiba oleoresin in a 10% cream formulation, and no healing potential in a 1% concentration [8]. However, cream formulations have low patient acceptance and less physical and chemical stability than emulgel formulations. Furthermore, nanostructured lipid carriers also increase the bioavailability and physical-chemical stability of drugs [9,10]. Therefore, our group synthesized an emulgel formulation containing nanostructured lipid carriers with 1% copaiba oleoresin (NLC) and confirmed the efficacy of the treatment in a rat skin excision wound model, increasing the healing effect of 1% copaiba oleoresin compared to the cream formulation of previous study [8]. Furthermore, the NLC formulation presented promising results compared to the three reference drugs. Although NeBa (antimicrobial) [2], Dex (cell proliferation) [3] and Col (extracellular matrix remodeling) [4] affect mainly one mechanism of wound healing, NLC treatment improves anti-inflammatory, re-epithelialization, wound retraction and remodeling mechanisms in all periods studied.

The retraction of wounds is mediated by myofibroblasts, cells that differentiate from fibroblasts stimulated by transforming growth factor-β1 (TGF-β1), acquiring a contractile phenotype due to intracellular proteins such as α-smooth muscle actin. Myofibroblasts bind to contractile proteins and multiple points of attachment, contracting them and reducing the area of the wounds [13,14]. Thus, we demonstrate the influence of the NLC formulation on the retraction mechanism, with better results compared to untreated lesions and reference drugs, improving wound retraction in the three studied periods. The retraction mechanism was mediated by the α-SMA pathway, increasing the retraction of the wounds due to increased numbers of immunolabeled myofibroblasts.

With the synthesis of a fibrin clot after the formation of a skin lesion, the cells release molecules that stimulate the migration of leukocytes to the wound. Firstly, neutrophils migrate to the region to promote the debridement of necrotic tissue and the phagocytosis of possible pathogens. After a few hours, macrophages migrate to the lesion, assisting in tissue debridement, antigen phagocytosis and the synthesis of cytokines and growth factors that will influence other healing mechanisms [15]. However, chronic leukocyte activity increases the production of ROS, causing local oxidative stress [16]. Therefore, the balance of antioxidant mechanisms and pro- and anti-inflammatory mediators is necessary for the continuity of the healing process, and understanding the role of these mediators in healing can lead to more efficient treatments for chronic injuries [17]. Moreover, analyzing the balance among pro- and anti-inflammatory mediators, several cytokines, such as IFN-γ, IL-1β, IL-6 and TNF-α, promote the differentiation, migration and activation of macrophages, neutrophils and NK cells and act on the synthesis of collagen by fibroblasts [18,19]. Therefore, the maintenance of pro-inflammatory cytokines at reduced levels suggested the anti-inflammatory potential of NLCs, with reduced concentrations of TNF-α, IL-1β, IL-6 and IFN-γ. Another result that suggests the anti-inflammatory effect of NLC formulations is the high concentration of IL-10. This is because IL-10 is an anti-inflammatory interleukin that inhibits the synthesis of pro-inflammatory mediators, preventing chronic inflammation and fibrosis during healing [18].

Reconstruction of the vascular networks of lesions is essential for the correct healing process, providing oxygen and nutrients necessary for the proliferation and migration of cells. However, an increase in the number of blood vessels over a long period of time can be considered one of the markers of tissue fibrosis, and after a period of time, endothelial cells undergo apoptosis, reducing vascularization to normal levels [16]. The data obtained by counting the number of blood vessels in the border and center of wounds did not show the effects of NLC on the angiogenesis mechanism.

Another important mechanism of skin wound healing, re-epithelialization, occurs with the proliferation of keratinocytes at the edges of the lesions, the dissolution of adhesion molecules and the synthesis of anchoring proteins to assist in migration through the extracellular matrix [20]. Desmoglein-3 is a transmembrane adhesion molecule in desmosomes that maintains keratinocyte attachment and is degraded in migratory keratinocytes during the re-epithelialization mechanism [20]. Laminin-γ2 is another essential protein with a role in re-epithelialization. It is synthesized by migratory keratinocytes from wound edges in the dermo-epidermal junction and assists in keratinocyte migration, anchoring the cells through the extracellular matrix [21]. However, the excessive proliferation of keratinocytes—as the result of chronic inflammation—can lead to an increase in the thickness of the epidermis. The keratinocytes from wound edges acquire a hyperproliferative, hyperkeratotic, and parakeratotic phenotype, with a delay in the migratory potential of these cells and impairment of the re-epithelialization mechanism, resulting in pathologic scars or non-healed wounds [22]. For this reason, it is necessary to control the proliferation and migration of keratinocytes for the correct re-epithelialization of the injury. Therefore, our immunohistochemistry results for Ki-67, desmoglein-3 and laminin-γ2 suggested the stimulation of treatments with NLC formulation for re-epithelialization, decreasing the amount of intercellular adhesion protein (desmoglein-3) and increasing the anchoring protein (laminin-γ2) levels during migration. Furthermore, we showed that although NLC treatment increased the migration of keratinocytes, there was no increase in the thickness of the epidermis as a sign of pathologic wound healing.

The last stage of skin wound healing is the remodeling of the extracellular matrix. In this mechanism, myofibroblasts, endothelial cells, and excess fibroblasts undergo apoptosis, type III collagen is degraded by extracellular matrix metalloproteinase, and the resistant and elastic permanent extracellular matrix (rich in collagen I and elastin) is synthesized [23]. However, remodeling of the extracellular matrix must be carefully controlled, because of chronic production of collagen I and mediators such as α-SMA and TGF-β1 can lead to chronic fibrosis. Thus, microscopic analysis of Mallory’s trichrome showed that NLC formulation influenced the synthesis of collagen in the early period of remodeling, anticipating the synthesis of collagen and the remodeling mechanism. Moreover, we showed a decrease in α-SMA immunolabeling NLC at 14 days, proving the absence of fibrosis after the first weeks as a consequence of the retraction mechanism.

## 4. Materials and Methods

### 4.1. Extraction of Oleoresin

The extraction of *Copaifera langsdorffii* oleoresin and its purification were reported by Ribeiro et al. (2019) (Figure 22) [24], and the characterization of the plant oleoresin was described by Souza et al. (2011) [25]. Briefly, samples of the *Copaifera langsdorffii* were collected in the north and southeast regions of Brazil and the plant voucher (SPFR 10120) of the plant was identified by Silvane Tavares Rodrigues at the Herbarium of EMBRAPRA—Oriental Amazônia, by direct comparison with authentic herbarium vouchers. The oleoresin was collected, and the volatile fraction was obtained by Victor Pena Ribeiro using spinning band distillation, and the chemical characterization of the oleoresin was carried out by gas chromatography equipped with a flame ionization detector.

### 4.2. Preparation and Characterization of Nanostructured Lipid Carriers with Copaifera langsdorffii Oleoresin

A nanostructured lipid carrier containing 1% of *Copaifera langsdorffii* oleoresin was made as described by Pivetta et al. (2018), with a few modifications [26]. The oil phase was prepared by mixing 200 mg of Illipe butter (Polytechno Indústrias Químicas Ltda, Guarulhos, Brazil) with *Copaifera langsdorffii* oleoresin (1% *w*/*w*) and the mixture was heated to 60 °C. An aqueous solution of Pluronic F68 (0.5% *w*/*v*) (Sigma-Aldrich, St. Louis, MO, USA) at 60 °C was added to the oil phase, followed by sonication (13 mm probe and 40% amplitude). Afterward, the dispersion was cooled to 25 °C, forming a nanostructured lipid carrier with oleoresin. The size, polydispersity index, and zeta potential of these nanostructures were determined by dynamic light scattering using a Zetasizer Nano ZS90 (Malvern Panalytical, Malvern, UK). The measurements were made with samples diluted in 1 mM KCl solution.

### 4.3. Preparation and Characterization of the Topical Formulation Containing NLC

The emulgel was made by homogenizing Sepineo P600 (3% *w*/*w*) (Spectrum Chemical, New Brunswick, NJ, USA), propylene glycol (5% *w*/*w*) (Spectrum Chemical, New Brunswick, USA), Labrafac lipophile WL 1349 (10% *w*/*w*) (Gattefossé, Lyon, France), methyl dibromo glutaronitrile/phenoxyethanol (0.1% *w*/*w*) (Spectrum Chemical, New Brunswick, USA) and water. Afterward, a *Copaifera langsdorffii* oleoresin-loaded nanostructured lipid carrier dispersion (30% *w*/*w*) was added to the emulgel and homogenized at 25 °C. The rheological behavior of the emulgel containing copaiba oleoresin-loaded nanostructured lipid carriers was analyzed in a Rheometer R/S plus (AMETEK Brookfield, Middleborough, MA, USA) equipped with a C50-1 spindle and RHEO Software 2000 version 2.8. The sample behavior was monitored at 25 °C using a water bath/circulator. The time of the upward curve was 120 s, with shear rates ranging from 0 to 1000 s^−1^, and 120 s to the downward curve, with shear rates ranging from 1000 to 0 s^−1^ [27].

### 4.4. Animals

Male Wistar rats (*Rattus norvegicus*) weighing 250 ± 20 g were used in the experiments. The animals were supplied by the Central Animal House, São Paulo State University (UNESP), Botucatu, and acclimated in individual cages under controlled conditions (23 ± 2 °C, 12 hs’ dark-light cycle, food and water ad libitum) until the experimental procedures were performed. All experiments were approved by the Ethics Committee on Animal Use (UNESP) under protocol 976/2017.

### 4.5. Excision Wound Model and Experimental Protocol

After the acclimation period, the animals were anesthetized with intraperitoneal ketamine (80 mg/kg) and xylazine (4 mg/kg), their dorsum was shaved, and a lesion was made in the dorsum (in the subscapular area) using a 3 cm diameter punch. The wound placed in this area could not be reached by the animals, which prevented self-licking [28]. Afterward, each rat was randomly distributed into six groups (n = 5/group): UT (wounded animal without treatment), NeBa (wounded animal treated with neomycin 5 mg/g + sulfate bacitracin zinc 250 IU/g), Dex (wounded animals treated with dexpanthenol 5%), Col (wounded animals treated with collagenase 1.2 IU), NLC (wounded animals treated with 1% *Copaifera langsdorffii* oleoresin loaded in nanostructured lipid carriers) and Control (animals without lesion and treatment—physiologic pattern). After the surgical procedure, the wounds of the UT, NeBa, Dex, Col and NLC groups were topically treated every day, twice a day, and the control group was used as the reference pattern of normal skin during three different experimental periods: 3, 7, or 14 days (30 animals/period). After each treatment period (3, 7, and 14 days), the animals were euthanized, and wound and blood samples were collected for biochemical, immunoenzymatic, immunohistochemical and histological analyses.

### 4.6. Macroscopic Analysis

To determine the macroscopic reduction in lesions, the wounded area was photographed every day using a scale bar during each experimental period. The wounded areas were measured using specific software, and the percentage of wound retraction was calculated according to the daily retraction of the lesions and compared to their initial size according to the formula:(1)$$\mathrm{Wound retraction}\;\left(\%\right)=\;\left(\frac{Initial\;wound\;area-Wound\;area\;in\;the\;analyzed\;day}{Initial\;wound\;area}\right)\times 100$$

### 4.7. Histopathological Analysis

The wound samples were fixed with 10% buffered formalin, processed in paraffin, sliced (5 µm thickness) and stained in HE and Mallory’s trichrome. HE staining was used to analyze the number of cells, epidermis thickness and number of blood vessels. Mallory’s trichrome was used to analyze the total amount of collagen and its deposition in the dermis. For each sample, the border and the center of the wounds were analyzed and photographed in five different fields (Appendix A). The measurements were made using AvSoft BioView 5.0 Spectra software.

### 4.8. Immunohistochemical Analysis

The wounded skin samples were fixed with 10% buffered formalin, processed in paraffin and sliced (5 µm thickness). The slices were submitted to antigen retrieval by pressure (125 °C/25 psi). Subsequently, the slices were processed according to the protocol of a mouse and rabbit-specific HRP/DAB detection kit micropolymer (Abcam, Cambridge, MA, USA) using primary antibodies against laminin-γ2 (1:200 µL) (Santa Cruz Biotechnology, Dallas, TX, USA), desmoglein-3 (1:200 µL) (Abcam, Cambridge, MA, USA), Ki-67 (1:100 µL) (Abcam, Cambridge, MA, USA) and α-smooth muscle actin (1:400 µL) (Abcam, Cambridge, MA, USA). For the analysis, all areas with immunolabeling corresponding to the primary antibodies were measured, and the positive cells were counted. The measurements were made using AvSoft BioView 5.0 Spectra software.

### 4.9. Enzyme-Inked Immunosorbent Assay (ELISA)

The skin samples were used to quantify cytokines. For this, interferon- γ (IFN-γ), interleukin-1β (IL-1β), interleukin-6 (IL-6) and tumor necrosis factor-α (TNF-α) were used as pro-inflammatory biomarkers, and interleukin-10 (IL-10) was used as an anti-inflammatory biomarker. First, the samples were homogenized in a 1:5 proportion (*m/v*) using phosphate saline buffer at pH 7.4 with protease inhibitor cocktail (1%). The homogenate was centrifuged for 15 min at 10,000 rpm and 4 °C. After centrifugation, the supernatant of each sample was collected sample and used to quantify the concentration of proteins using the Bradford experiment [29] and each cytokine as described by the protocols of each kit (R&D Systems, Minneapolis, MN, USA). The results were expressed in pg of cytokine/mg of protein.

### 4.10. Toxicological Analysis

The blood of each animal was collected immediately after euthanasia and centrifuged for 15 min at 6000 rpm and 4 °C. The parameters of liver and kidney toxicity were evaluated through quantification of aspartate aminotransferase (AST), alanine aminotransferase (ALT) and γ-glutamyl transferase (γ-GT) activities (IU/L) and the concentrations of urea and creatinine (mg/dL) as described by the protocols of specific kits (Interteck-Katal, Belo Horizonte, Brazil).

### 4.11. Statistical Analysis

Parametric data were subjected to one-way ANOVA and Tukey’s post hoc test. Non-parametric results were subjected to the Kruskal-Wallis test, followed by the Dunn posttest. The analyses were performed by GraphPad Prism 5.0 software, with *p* < 0.05.

## 5. Conclusions

With our results, we suggested that a nanostructured lipid carrier containing oleoresin of *Copaifera langsdorffii* at 1% improved skin wound healing in a rat excision model through anti-inflammatory activity, retraction of the lesion mediated by α-smooth muscle actin and re-epithelialization through desmoglein-3 and laminin-γ2 mechanisms. Moreover, the use of nanostructured lipid carriers improved the healing activity of 1% copaiba oleoresin, compared to the ineffectiveness of 1% oleoresin cream formulation. Through the analysis of liver enzymes and kidney proteins, we demonstrated the safety of the topic use of NLC, without systemic toxicity. Furthermore, we showed the promising potential of the NLC compared to three different commercial drugs used to treat wounds, with the improvement of essential mechanisms of wound healing by NLC. Therefore, we believe that this formulation has great potential in the treatment of skin wounds in the future with further investigation.

## Figures and Tables

**Figure 1 ijms-24-15882-f001:**
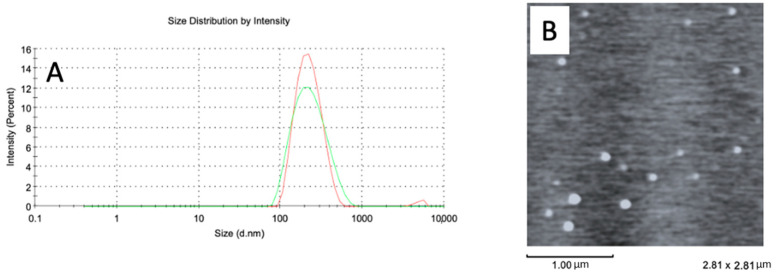
(**A**) Size distribution profile of NLC with *Copaifera langsdorffii* oleoresin after preparation (red curve) and after 60 days (green curve), (**B**) AFM image of NLC with *Copaifera langsdorffii* oleoresin. Scale bar/magnification of 1 µm.

**Figure 2 ijms-24-15882-f002:**
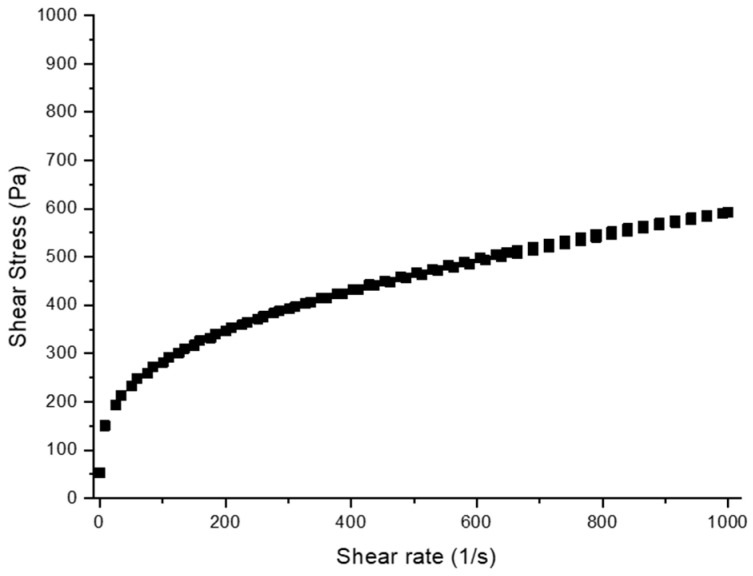
Shear rate-shear stress plot of the topical formulation containing NLC with *Copaifera langsdorffii* oleoresin.

**Figure 3 ijms-24-15882-f003:**
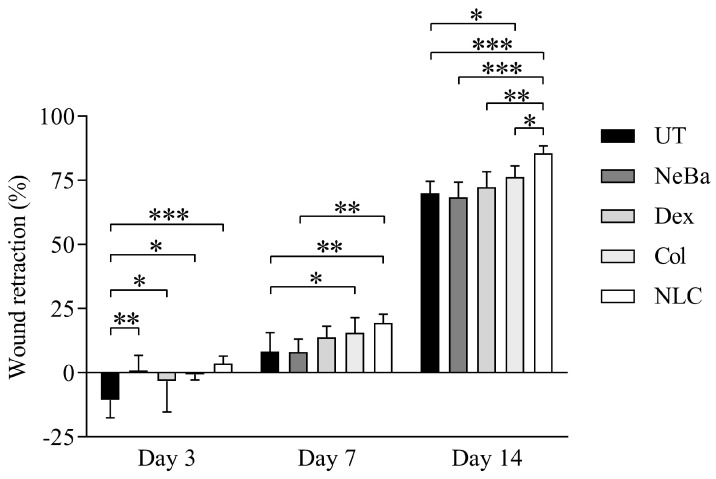
Effect of different treatments on the healing of skin wounds during 3, 7 and 14 days. * *p* < 0.05, ** *p* < 0.01, *** *p* < 0.001 according to one-way ANOVA and Tukey test for each period of treatment (n = 5/group). UT: wounded animals without treatment. NeBa: wounded animals treated with neomycin + sulfate bacitracin zinc. Dex: wounded animals treated with dexpanthenol. Col: wounded animals treated with collagenase. NLC: wounded animals treated with 1% *Copaifera langsdorffii* oleoresin loaded in nanostructured lipid carriers.

**Figure 4 ijms-24-15882-f004:**
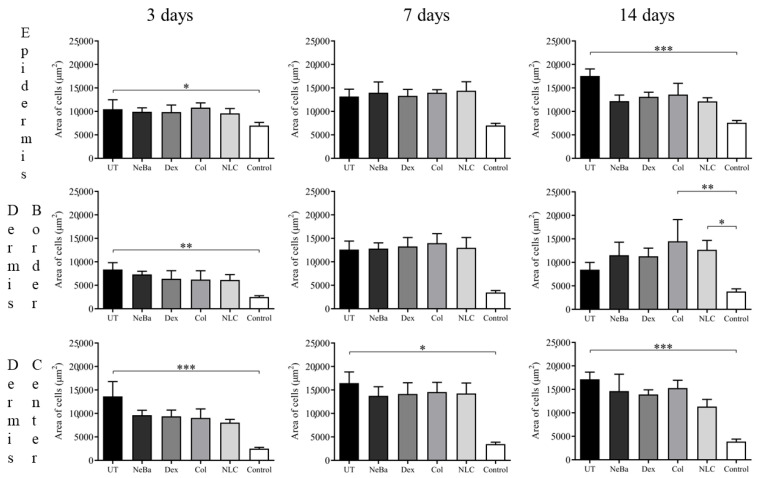
Quantification of the area (µm^2^) of cells in the epidermis, border and center of the dermis of cutaneous wounds of UT, NeBa, Dex, Col, NLC and Control groups during 3, 7 and 14 days. * *p* < 0.05, ** *p* < 0.01, *** *p* < 0.001 according to the Kruskal–Wallis test, followed by Dunn post-test (n = 5/group). UT: wounded animals without treatment. NeBa: wounded animals treated with neomycin + sulfate bacitracin zinc. Dex: wounded animals treated with dexpanthenol. Col: wounded animals treated with collagenase. NLC: wounded animals treated with 1% *Copaifera langsdorffii* oleoresin loaded in nanostructured lipid carriers. Control: animals without lesions and treatment.

**Figure 5 ijms-24-15882-f005:**
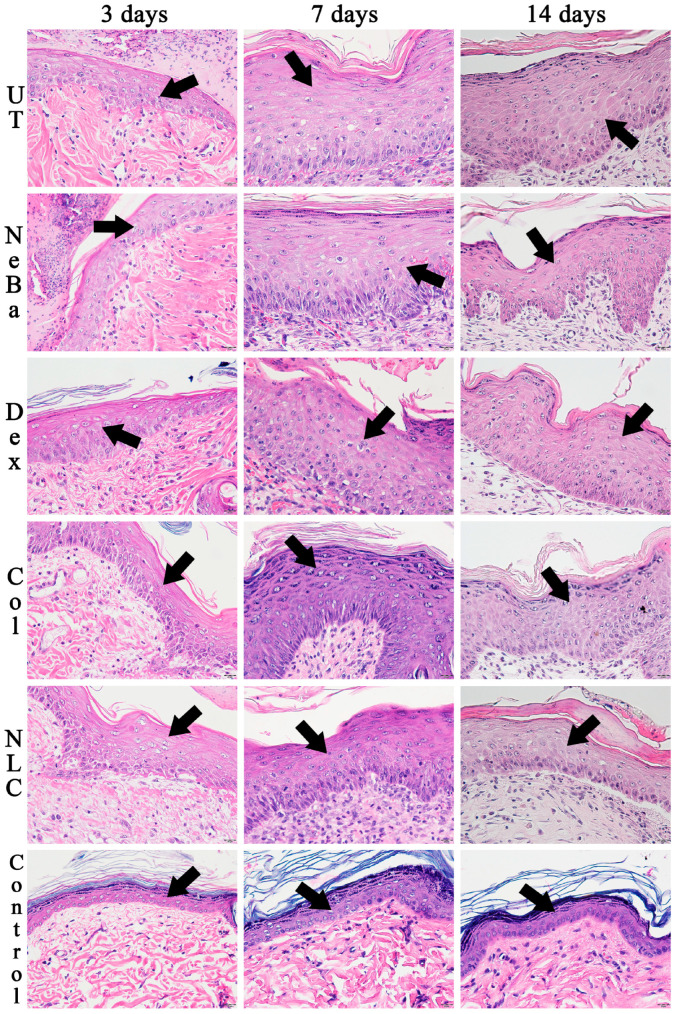
Hematoxylin and eosin photomicrographs of the epidermis in UT, NeBa, Dex, Col, NLC and Control groups during 3, 7 and 14 days. Black arrows of each photomicrography indicate the area of epidermis labeled purple. UT: wounded animal without treatment. NeBa: wounded animal treated with neomycin + sulfate bacitracin zinc. Dex: wounded animals treated with dexpanthenol. Col: wounded animals treated with collagenase. NLC: wounded animals treated with 1% *Copaifera langsdorffii* oleoresin loaded in nanostructured lipid carriers. Control: animals without lesions and treatment. Scale bar in each figure represents 20 µm.

**Figure 6 ijms-24-15882-f006:**
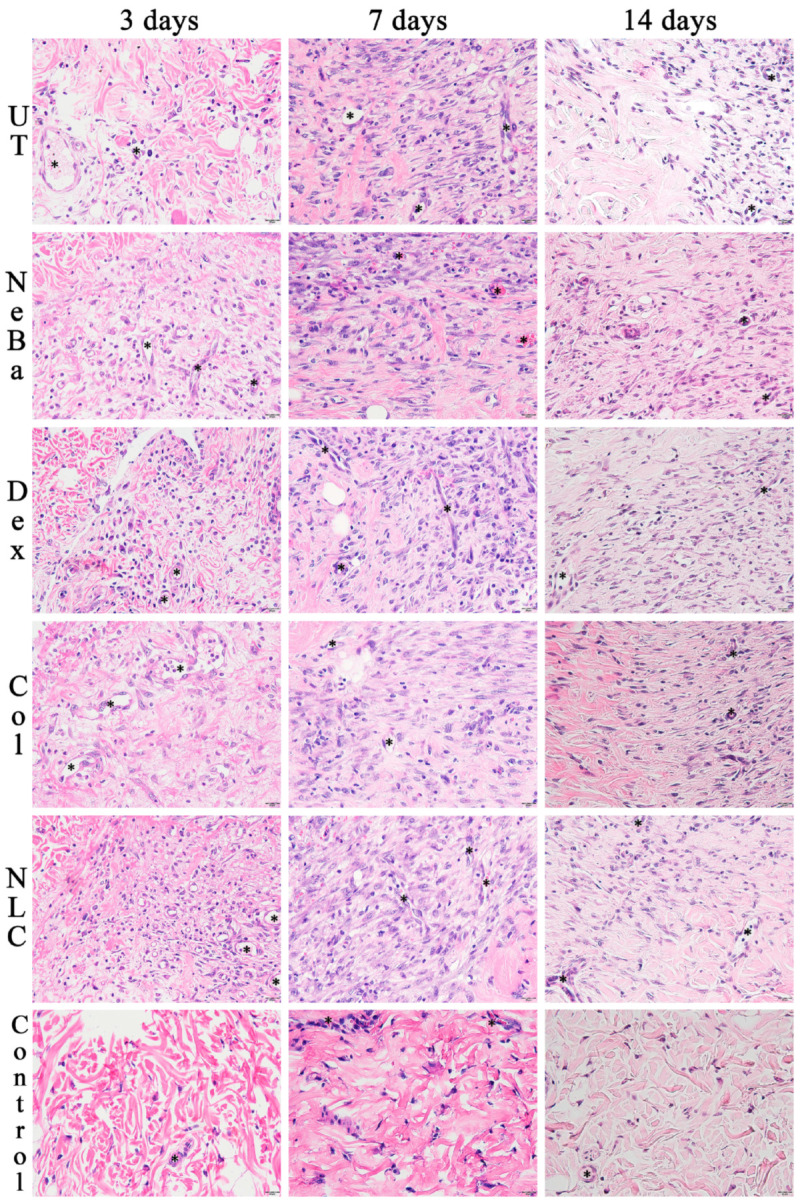
Hematoxylin and eosin photomicrographs of the border of wounds in the dermis of UT, NeBa, Dex, Col, NLC and Control groups during 3, 7 and 14 days. The area of each photomicrography represents the border of the wound (transition between the normal skin and the central region of the wound). The cells were stained purple and the extracellular matrix was stained pink. Black asterisks indicate the blood vessels. UT: wounded animal without treatment. NeBa: wounded animal treated with neomycin + sulfate bacitracin zinc. Dex: wounded animals treated with dexpanthenol. Col: wounded animals treated with collagenase. NLC: wounded animals treated with 1% *Copaifera langsdorffii* oleoresin loaded in nanostructured lipid carriers. Control: animals without lesion and treatment. Scale bar in each figure represents 20 µm.

**Figure 7 ijms-24-15882-f007:**
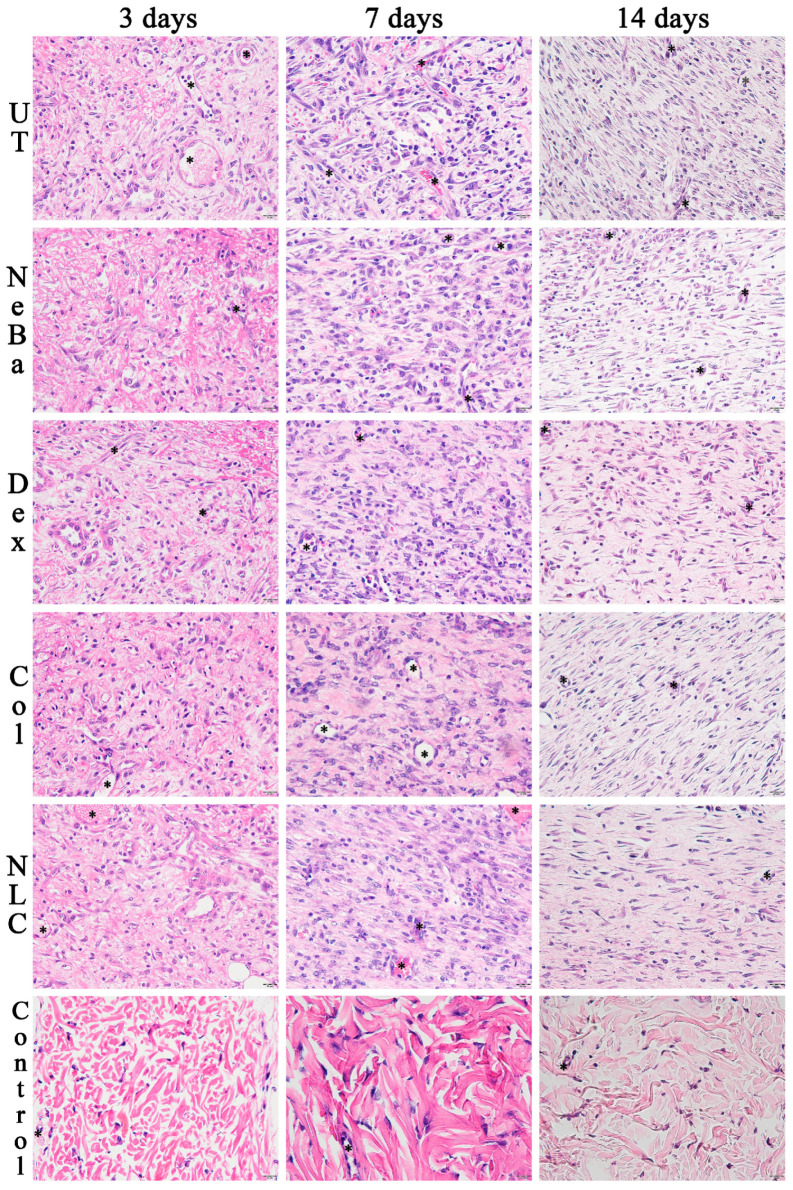
Hematoxylin and eosin photomicrographs of the center of wounds in the dermis of UT, NeBa, Dex, Col, NLC and Control groups during 3, 7 and 14 days. The area of each photomicrography represents the central region of the wound. The cells were stained purple, and the extracellular matrix was stained pink. Black asterisks indicate the blood vessels. UT: wounded animal without treatment. NeBa: wounded animal treated with neomycin + sulfate bacitracin zinc. Dex: wounded animals treated with dexpanthenol. Col: wounded animals treated with collagenase. NLC: wounded animals treated with 1% *Copaifera langsdorffii* oleoresin loaded in nanostructured lipid carriers. Control: animals without lesions and treatment. Scale bar in each figure represents 20 µm.

**Figure 8 ijms-24-15882-f008:**
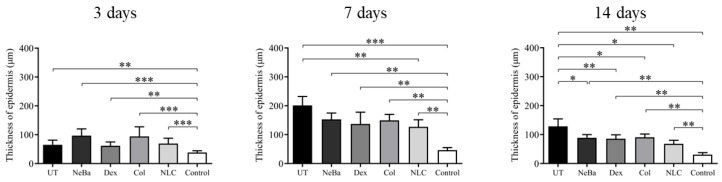
Measurement of epidermis thickness (µm) of cutaneous wounds treated with NeBa, Dex, Col or NLC during 3, 7 and 14 days. * *p* < 0.05, ** *p* < 0.01, *** *p* < 0.001 according to the Kruskal–Wallis test, followed by Dunn post-test (n = 5/group). UT: wounded animals without treatment. NeBa: wounded animals treated with neomycin + sulfate bacitracin zinc. Dex: wounded animals treated with dexpanthenol. Col: wounded animals treated with collagenase. NLC: wounded animals treated with 1% *Copaifera langsdorffii* oleoresin loaded in nanostructured lipid carriers. Control: animals without lesions and treatment.

**Figure 9 ijms-24-15882-f009:**
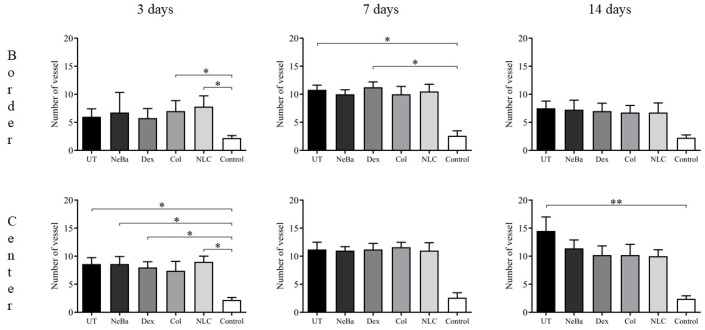
Measurement of blood vessels in the border and center of the dermis in wounds treated with NeBa, Dex, Col or NLC during 3, 7 and 14 days. * *p* < 0.05, ** *p* < 0.01 according to the Kruskal–Wallis test, followed by Dunn post-test (n = 5/group). UT: wounded animals without treatment. NeBa: wounded animals treated with neomycin + sulfate bacitracin zinc. Dex: wounded animals treated with dexpanthenol. Col: wounded animals treated with collagenase. NLC: wounded animals treated with 1% *Copaifera langsdorffii* oleoresin loaded in nanostructured lipid carriers. Control: animals without lesions and treatment.

**Figure 10 ijms-24-15882-f010:**
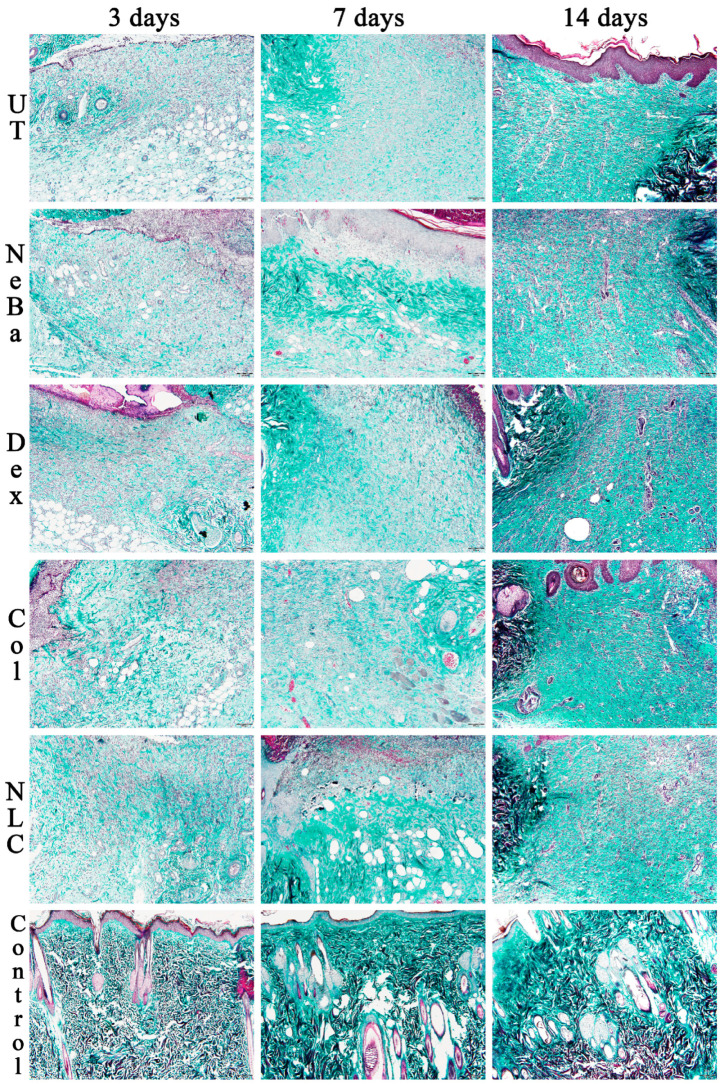
Mallory’s trichrome photomicrographs of the border of wounds in the dermis of UT, NeBa, Dex, Col, NLC and Control groups during 3, 7 and 14 days. UT: wounded animal without treatment. NeBa: wounded animal treated with neomycin + sulfate bacitracin zinc. Dex: wounded animals treated with dexpanthenol. Col: wounded animals treated with collagenase. NLC: wounded animals treated with 1% *Copaifera langsdorffii* oleoresin loaded in nanostructured lipid carriers. Control: animals without lesions and treatment. The greenish stain represents the area of collagen content in the border of dermis. Scale bar in each figure represents 20 µm.

**Figure 11 ijms-24-15882-f011:**
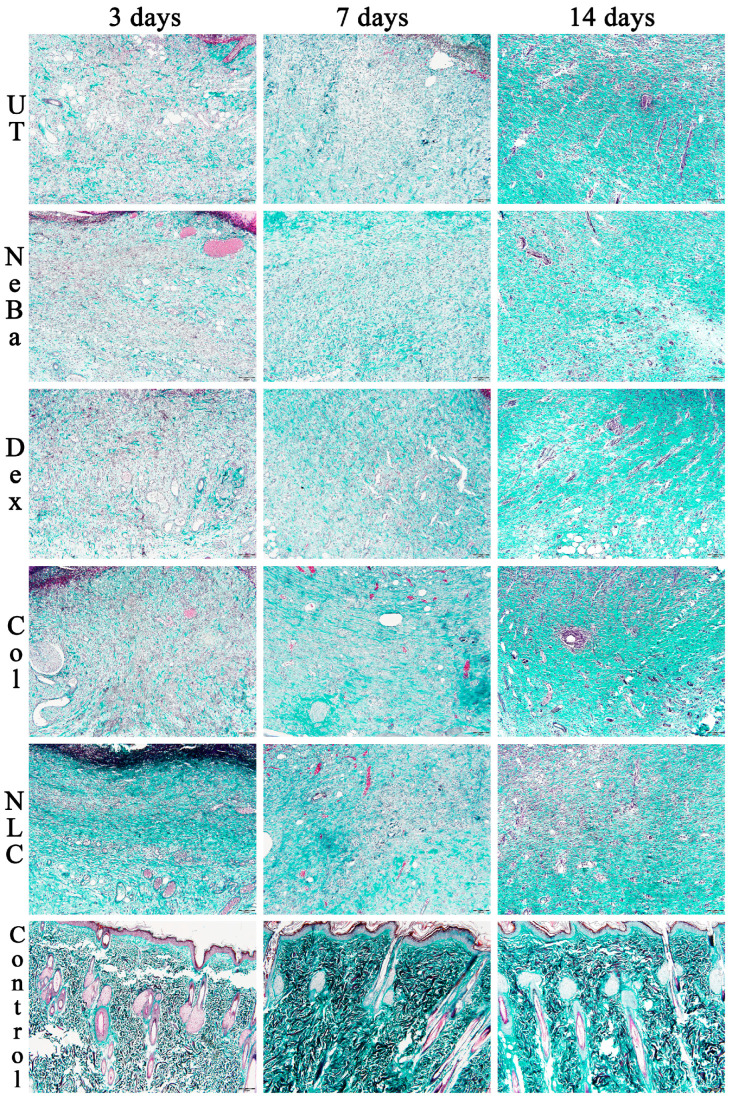
Mallory’s trichrome photomicrographs of the center of wounds in the dermis of UT, NeBa, Dex, Col, NLC and Control groups during 3, 7 and 14 days. UT: wounded animal without treatment. NeBa: wounded animal treated with neomycin + sulfate bacitracin zinc. Dex: wounded animals treated with dexpanthenol. Col: wounded animals treated with collagenase. NLC: wounded animals treated with 1% *Copaifera langsdorffii* oleoresin loaded in nanostructured lipid carriers. Control: animals without lesions and treatment. The greenish stain represents the area of collagen content in the center of dermis. Scale bar in each figure represents 20 µm.

**Figure 12 ijms-24-15882-f012:**
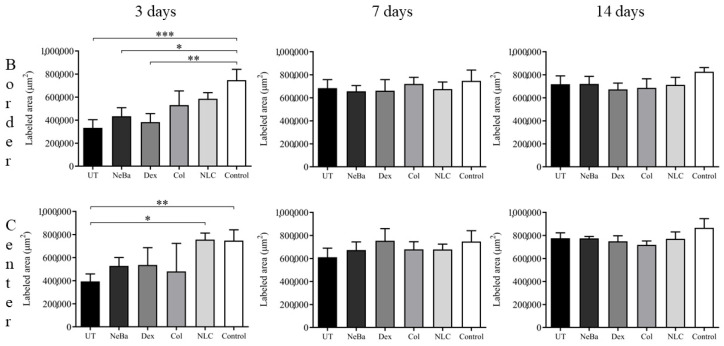
Quantification of collagen area (µm^2^) at the border and center of the dermis in wounds of UT, NeBa, Dex, Col, NLC and Control groups during 3, 7 and 14 days. * *p* < 0.05, ** *p* < 0.01, *** *p* < 0.001 according to the Kruskal-Wallis test, followed by Dunn post-test (n = 5/group). UT: wounded animals without treatment. NeBa: wounded animals treated with neomycin + sulfate bacitracin zinc. Dex: wounded animals treated with dexpanthenol. Col: wounded animals treated with collagenase. NLC: wounded animals treated with 1% *Copaifera langsdorffii* oleoresin loaded in nanostructured lipid carriers. Control: animals without lesions and treatment.

**Figure 13 ijms-24-15882-f013:**
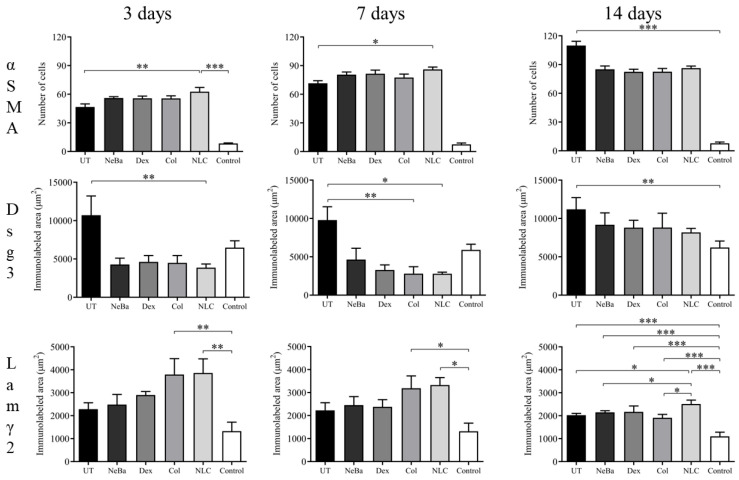
Immunolabeled area of desmoglein-3 (Dsg3) (µm^2^), laminin-γ2 (Lamγ2) (µm^2^), and number of positive α-SMA cells in skin wounds of different treatments during 3, 7 and 14 days. * *p* < 0.05, ** *p* < 0.01, *** *p* < 0.001 according to one-way ANOVA, followed by Tukey test (n = 5/group). UT: wounded animal without treatment. NeBa: wounded animal treated with neomycin + sulfate bacitracin zinc. Dex: wounded animals treated with dexpanthenol. Col: wounded animals treated with collagenase. NLC: wounded animals treated with 1% *Copaifera langsdorffii* oleoresin loaded in nanostructured lipid carriers. Control: animals without lesions and treatment.

**Figure 14 ijms-24-15882-f014:**
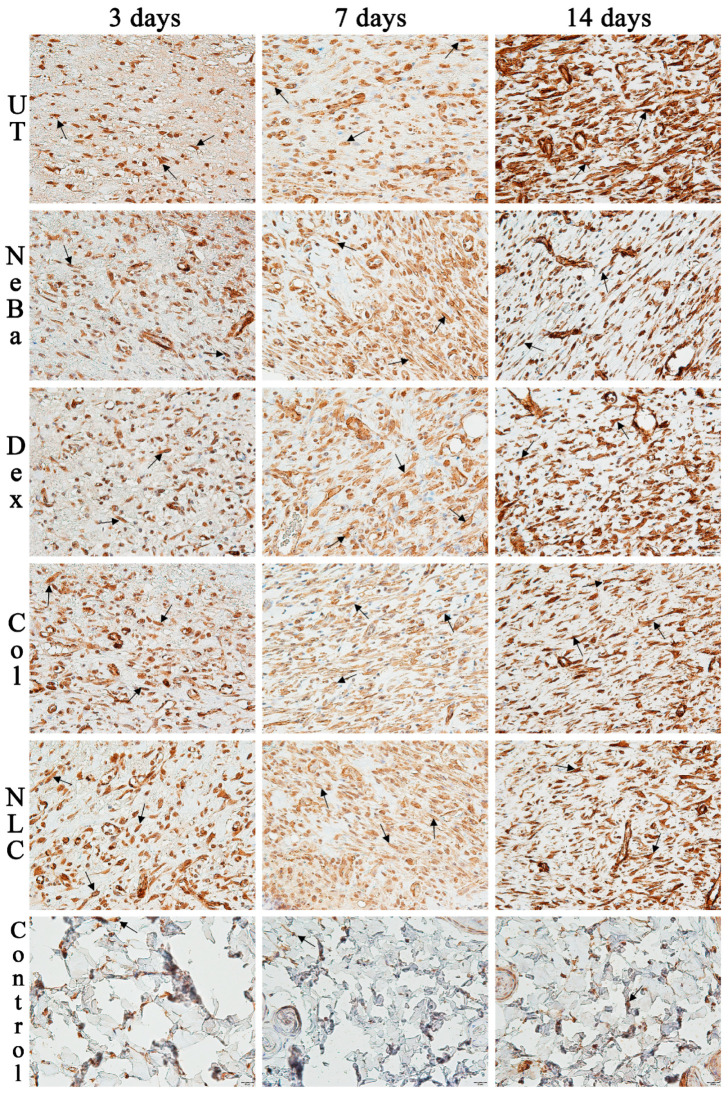
Photomicrographs of the immunolabeling of α-SMA in the dermis of wounds treated during 3, 7 and 14 days. Black arrows indicate the immunolabeled myofibroblasts (brown color) positive for α-SMA presented in the center of the dermis. The remaining tissue was counterstained with hematoxylin (blue-purple color). UT: wounded animal without treatment. NeBa: wounded animal treated with neomycin + sulfate bacitracin zinc. Dex: wounded animals treated with dexpanthenol. Col: wounded animals treated with collagenase. NLC: wounded animals treated with 1% *Copaifera langsdorffii* oleoresin loaded in nanostructured lipid carriers. Control: animals without lesions and treatment. Scale bar in each figure represents 20 µm.

**Figure 15 ijms-24-15882-f015:**
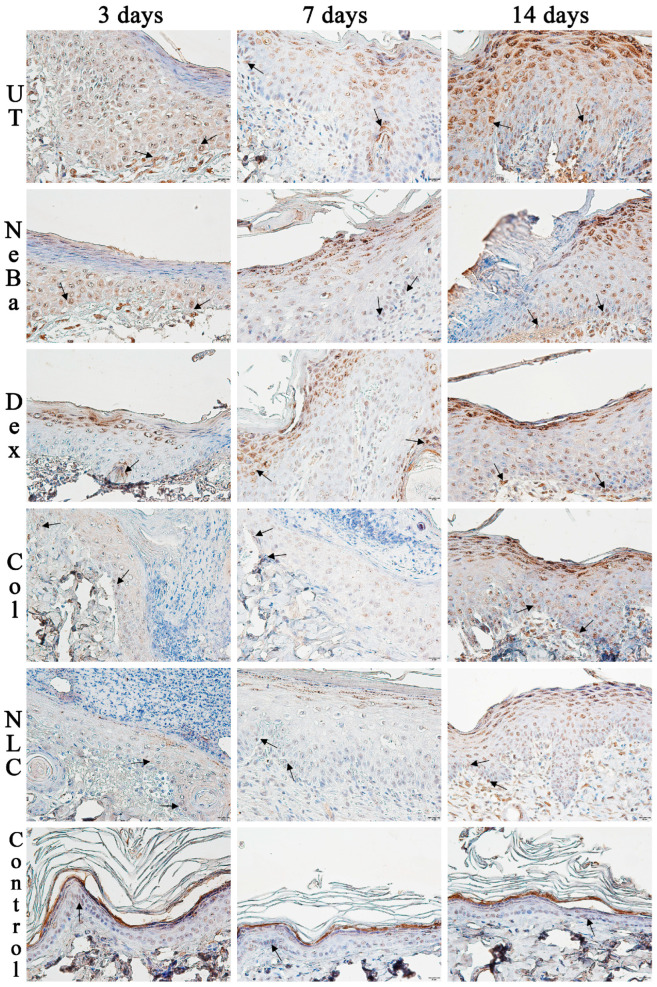
Photomicrographs of the immunolabeling of desmoglein-3 in the epidermis of wounds treated during 3, 7 and 14 days. Black arrows indicate the positive desmoglein-3 immunolabeled area (brown color) in the epidermis of the wound edges. The remaining tissue was counterstained with hematoxylin (blue-purple color). UT: wounded animal without treatment. NeBa: wounded animal treated with neomycin + sulfate bacitracin zinc. Dex: wounded animals treated with dexpanthenol. Col: wounded animals treated with collagenase. NLC: wounded animals treated with 1% *Copaifera langsdorffii* oleoresin loaded in nanostructured lipid carriers. Control: animals without lesions and treatment. Scale bar in each figure represents 20 µm.

**Figure 16 ijms-24-15882-f016:**
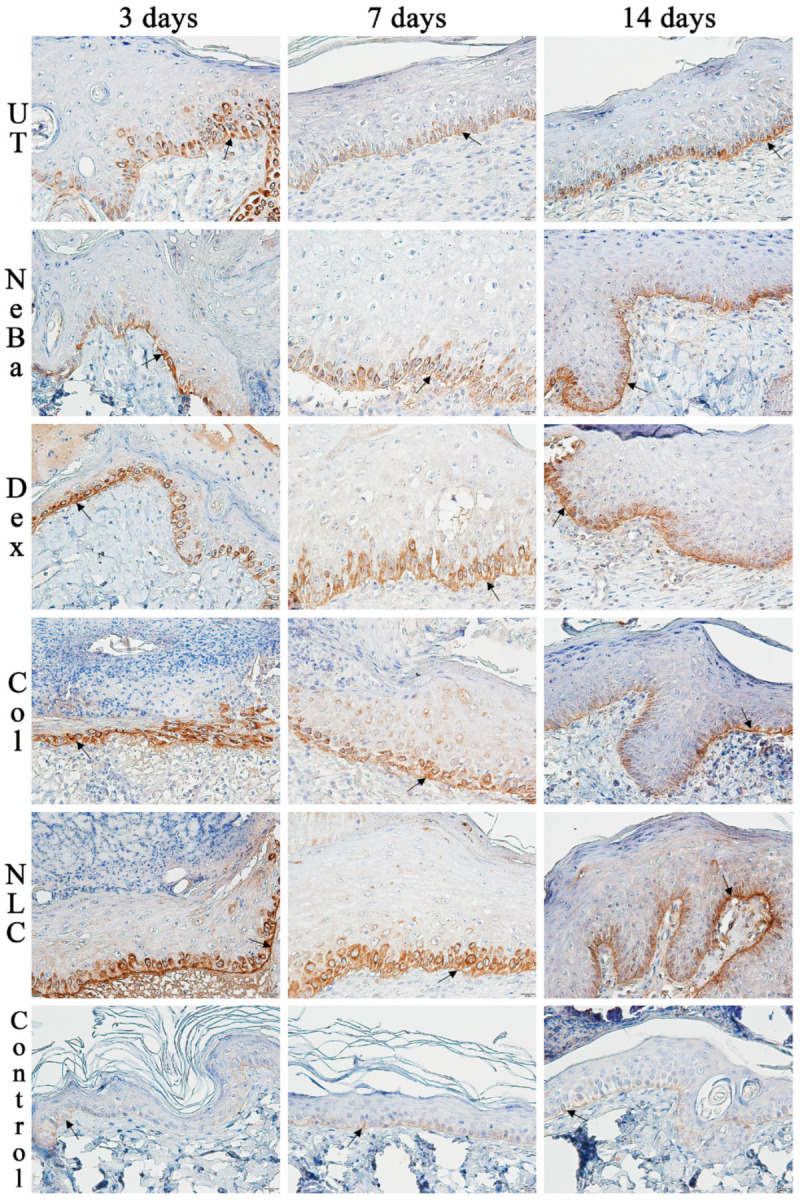
Photomicrographs of the immunolabeling of laminin-γ2 in the epidermis of wounds treated during 3, 7 and 14 days. Black arrows indicate the positive laminin-γ2 immunolabeled area (brown color) in the epidermis of the wound edges. The remaining tissue was counterstained with hematoxylin (blue-purple color). UT: wounded animal without treatment. NeBa: wounded animal treated with neomycin + sulfate bacitracin zinc. Dex: wounded animals treated with dexpanthenol. Col: wounded animals treated with collagenase. NLC: wounded animals treated with 1% *Copaifera langsdorffii* oleoresin loaded in nanostructured lipid carriers. Control: animals without lesions and treatment. Scale bar in each figure represents 20 µm.

**Figure 17 ijms-24-15882-f017:**
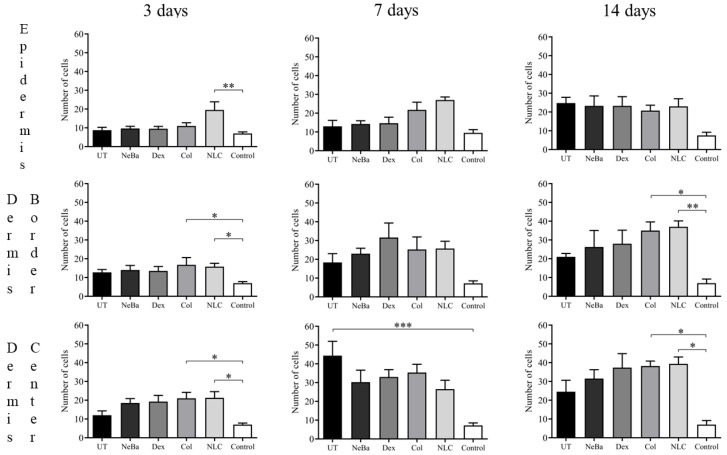
Ki-67 immunolabeling quantification of proliferating cells in the epidermis, border and center of the dermis in wounds treated during 3, 7 and 14 days. * *p* < 0.05, ** *p* < 0.01, *** *p* < 0.001 according to the Kruskal–Wallis test, followed by the Dunn test (n = 5/group). UT: wounded animal without treatment. NeBa: wounded animal treated with neomycin + sulfate bacitracin zinc. Dex: wounded animals treated with dexpanthenol. Col: wounded animals treated with collagenase. NLC: wounded animals treated with 1% *Copaifera langsdorffii* oleoresin loaded in nanostructured lipid carriers. Control: animals without lesions and treatment.

**Figure 18 ijms-24-15882-f018:**
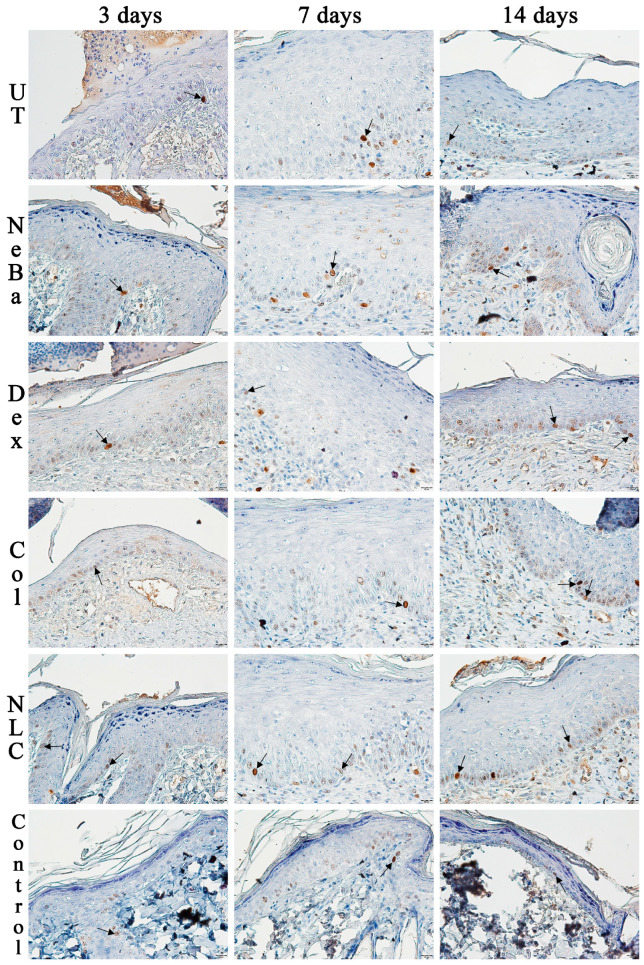
Photomicrographs of Ki-67 immunolabeling in the epidermis of wounds treated during 3, 7 and 14 days. Arrows indicate the positive Ki-67 immunolabeled cells. UT: wounded animal without treatment. NeBa: wounded animal treated with neomycin + sulfate bacitracin zinc. Dex: wounded animals treated with dexpanthenol. Col: wounded animals treated with collagenase. NLC: wounded animals treated with 1% *Copaifera langsdorffii* oleoresin loaded in nanostructured lipid carriers. Control: animals without lesions and treatment. Scale bar in each figure represents 20 µm.

**Figure 19 ijms-24-15882-f019:**
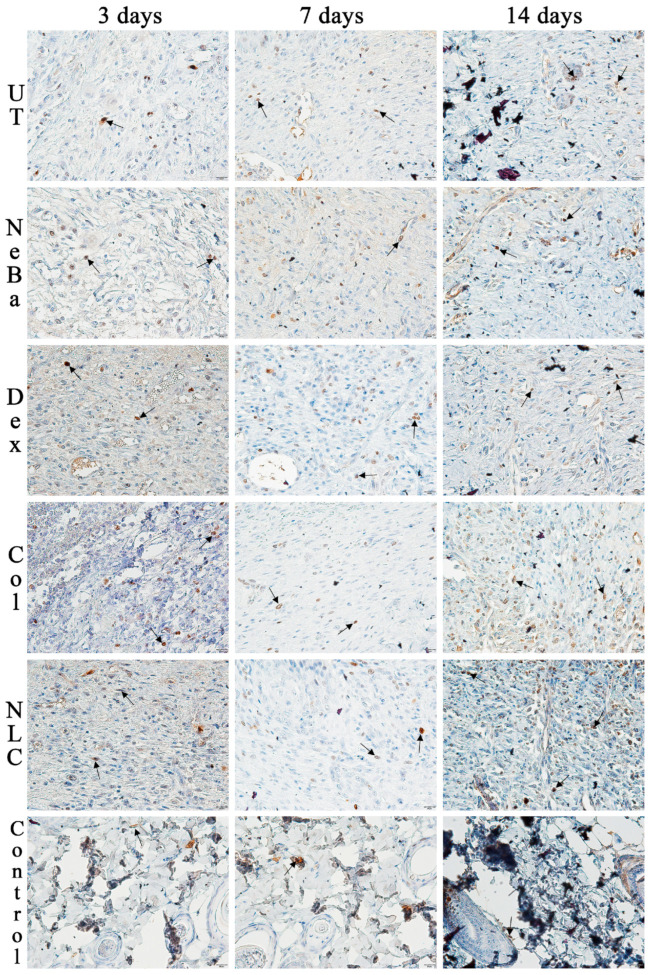
Photomicrographs of Ki-67 immunolabeling of the border of the wounds in the dermis of wounds treated during 3, 7 and 14 days. Arrows indicate the positive Ki-67 immunolabeled cells. UT: wounded animal without treatment. NeBa: wounded animal treated with neomycin + sulfate bacitracin zinc. Dex: wounded animals treated with dexpanthenol. Col: wounded animals treated with collagenase. NLC: wounded animals treated with 1% *Copaifera langsdorffii* oleoresin loaded in nanostructured lipid carriers. Control: animals without lesions and treatment. Scale bar in each figure represents 20 µm.

**Figure 20 ijms-24-15882-f020:**
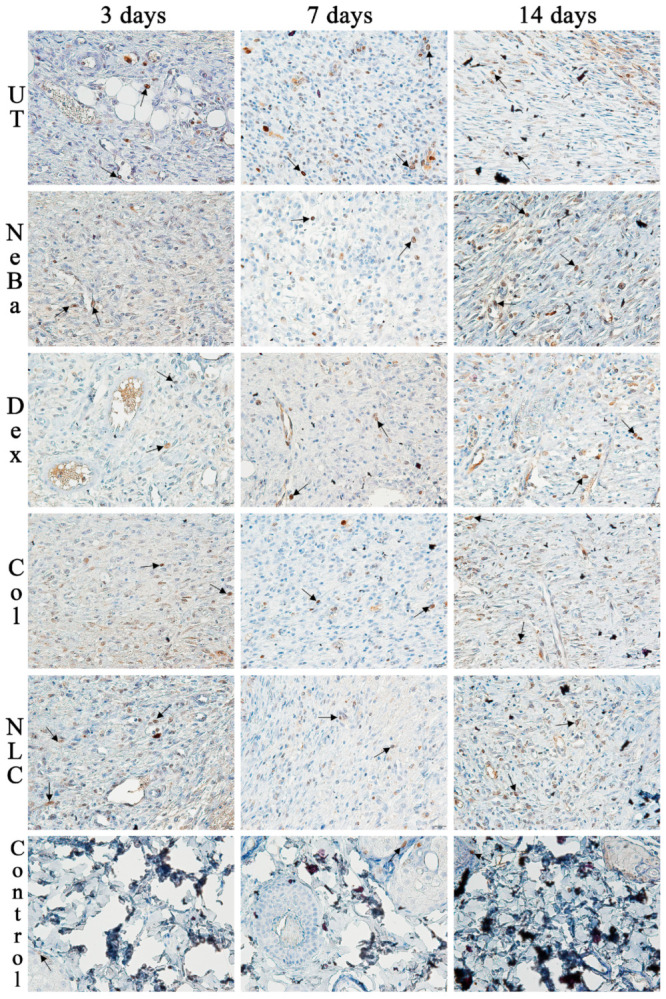
Photomicrographs of Ki-67 immunolabeling of the center of the wounds in the dermis of wounds treated during 3, 7 and 14 days. Arrows indicate the positive Ki-67 immunolabeled cells. UT: wounded animal without treatment. NeBa: wounded animal treated with neomycin + sulfate bacitracin zinc. Dex: wounded animals treated with dexpanthenol. Col: wounded animals treated with collagenase. NLC: wounded animals treated with 1% *Copaifera langsdorffii* oleoresin loaded in nanostructured lipid carriers. Control: animals without lesions and treatment. Scale bar in each figure represents 20 µm.

**Figure 21 ijms-24-15882-f021:**
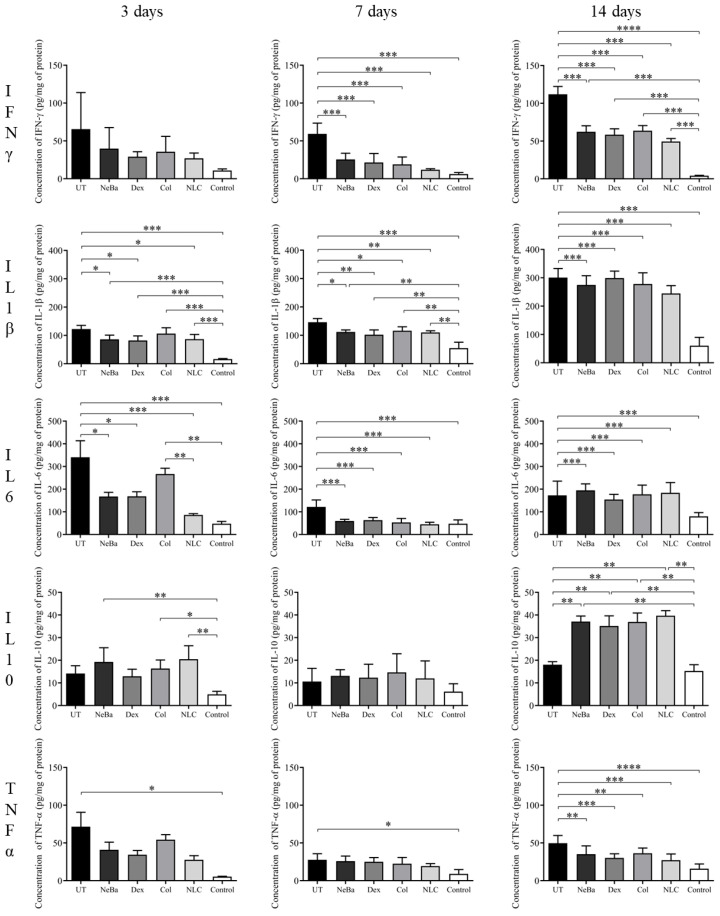
Concentrations of IFN-γ, IL-1β, IL-6, IL-10 and TNF-α (pg/mg protein) in skin wounds of rats treated for 3, 7 and 14 days. * *p* < 0.05, ** *p* < 0.01, *** *p* < 0.001, **** *p* < 0.0001 according to one-way ANOVA followed by Tukey test (n = 5/group). UT: wounded animal without treatment. NeBa: wounded animal treated with neomycin + sulfate bacitracin zinc. Dex: wounded animals treated with dexpanthenol. Col: wounded animals treated with collagenase. NLC: wounded animals treated with 1% *Copaifera langsdorffii* oleoresin loaded in nanostructured lipid carriers. Control: animals without lesions and treatment.

**Figure 22 ijms-24-15882-f022:**
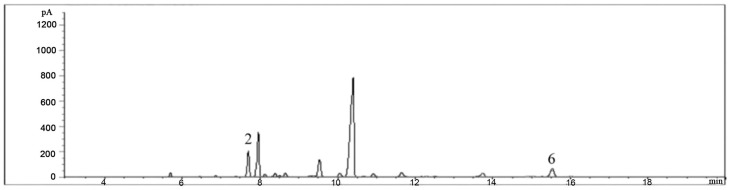
Gas Chromatography profile of the volatile oleoresin of the *Copaifera langsdorffii*. 2: β-caryophyllene; 6: caryophyllene oxide. Figure adapted with permission from Ribeiro et al. (2019) [24].

**Table 1 ijms-24-15882-t001:** Systemic toxicity analysis data for liver (AST, ALT, *γ*-GT) and renal (creatinine, urea) parameters in the serum of rats treated for 14 days.

Groups	AST (IU/L)	ALT (IU/L)	γ-GT (IU/L)	Creatinine (mg/dL)	Urea (mg/dL)
Untreated	143 ± 28	62 ± 12	1.2 ± 0.4	0.30 ± 0.03	44 ± 2.1
NeBa	138 ± 17	65 ± 7.7	1.1 ± 0.3	0.31 ± 0.04	44 ± 6.2
Dex	150 ± 28	80 ± 17	0.9 ± 0.3	0.27 ± 0.03	43 ± 5.6
Col	153 ± 29	63 ± 12	1.0 ± 0.2	0.30 ± 0.05	45 ± 4.6
NLC	147 ± 49	67 ± 13	1.0 ± 0.2	0.27 ± 0.04	45 ± 5.5
Control	147 ± 25	65 ± 9.1	1.0 ± 0.2	0.30 ± 0.02	44 ± 4.9

Untreated: wounded animal without treatment. NeBa: wounded animal treated with neomycin + sulfate bacitracin zinc. Dex: wounded animals treated with dexpanthenol. Col: wounded animals treated with collagenase. NLC: wounded animals treated with 1% *Copaifera langsdorffii* oleoresin loaded in nanostructured lipid carriers. Control: animals without lesions and treatment. AST: aspartate aminotransferase. ALT: alanine aminotransferase. γ-GT: γ-glutamyl transferase.

## Data Availability

All data presented in this article can be found in the Institutional Repository of São Paulo State University, available in: http://hdl.handle.net/11449/214352 (accessed on 30 June 2023).

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
