# Peer review of "Copaifera langsdorffii Oleoresin-Loaded Nanostructured Lipid Carrier Emulgel Improves Cutaneous Healing by Anti-Inflammatory and Re-Epithelialization Mechanisms"

_ijms, 2023, doi:10.3390/ijms242115882_

Round 1

Reviewer 1 Report

Comments and Suggestions for Authors

Lucas F. S. Gushiken et al reported research manuscript titled "Copaifera langsdorffii oleoresin-loaded nanostructured lipid carrier emulgel improves cutaneous healing by anti-inflammatory and re-epithelialization mechanisms". Authors developed a nanostructed lipid carrier formulation based on Copaifera langsdorffii 1% oleoresin. Compared the wound healing potential with commercially available formulations. Authors carried out histopathological, immunohistochemical analysis, ELISA, and toxicology studies on rats. Results showed that the rats treated with nanostructured lipd carrier (NLC) Copaifera langsdorffii 1% oleoresin formulation showed effective wound retraction, anti-inflammatory activity, and improved re-epithelialization. 

Overall, the research was well conducted and fits the scope and aim of this journal.

Comments:

Since this entire research is based on nano formulation. Authors need to provide accurate size of the formulation. Electron microscopy (TEM or SEM) studies will give you the actual size of the nanomaterials. 

Zeta potential of the nano dispersion found to be around -17 mV can precipitate out the nanoparticles from the dispersion ultimately leading to less stable formulation. Authors should provide comments/experiments to improve the stability of the nanoparticles in the dispersion.

Figures for the zeta size, potential, and rheological properties of the formulation should be provided in the supporting information.

Author Response

Dear reviewer,

On behalf of the entire team involved in the production of this manuscript, I would like to thank you for your comments and suggestions that contributed to improving the quality of our manuscript.

Below are the responses to the suggestions of our manuscript.

I remain at your disposal for any questions and clarifications.

Best regards,

Lucas Fernando Sérgio Gushiken

Comments:

Lucas F. S. Gushiken et al reported a research manuscript titled "Copaifera langsdorffii oleoresin-loaded nanostructured lipid carrier emulgel improves cutaneous healing by anti-inflammatory and re-epithelialization mechanisms". The authors developed a nanostructed lipid carrier formulation based on Copaifera langsdorffii 1% oleoresin. Compared the wound healing potential with commercially available formulations. Authors carried out the histopathological, immunohistochemical analysis, ELISA, and toxicology studies on rats. Results showed that the rats treated with nanostructured lipd carrier (NLC) Copaifera langsdorffii 1% oleoresin formulation showed effective wound retraction, anti-inflammatory activity, and improved re-epithelialization. 

Overall, the research was well conducted and fits the scope and aim of this journal.

Since this entire research is based on nano formulation. Authors need to provide accurate size of the formulation. Electron microscopy (TEM or SEM) studies will give you the actual size of the nanomaterials. Zeta potential of the nano dispersion found to be around -17 mV can precipitate out the nanoparticles from the dispersion ultimately leading to less stable formulation. Authors should provide comments/experiments to improve the stability of the nanoparticles in the dispersion.

Figures for the zeta size, potential, and rheological properties of the formulation should be provided in the supporting information.

Answer: Firstly, I would like to thank the reviewer for the comments that helped to improve our article. Regarding the microscopy images, the atomic force microscopy was included in the revised manuscript. The NLC-oleoresin stability can be attributed to the steric effect resulting from the use of the non-ionic surfactant Pluronic F68, in conjunction with electrostatic stability due to negative zeta potential. In this case, there is no specific zeta potential value to ensure the stability of the nanoparticle (Cortes et al., 2021). Therefore, the stability should be evaluated over time. Because of this, we evaluated the stability regarding the size and polydispersity index for 60 days. This information and discussion were added to the revised manuscript. The data of the size of nanoparticles after its preparation and after 60 days was added to the revised manuscript. The rheological properties of topical formulation were already discussed in the manuscript.

References:

Cortés, H.; Hernández-Parra, H.; Bernal-Chávez, S.A.; Prado-Audelo, M.L.D.; Caballero-Florán, I.H.; Borbolla-Jiménez, F.V.; González-Torres, M.; Magaña, J.J.; Leyva-Gómez, G. Non-Ionic Surfactants for Stabilization of Polymeric Nanoparticles for Biomedical Uses. Materials 2021, 14; 3197. doi: 10.3390/ma14123197.

Reviewer 2 Report

Comments and Suggestions for Authors

Dear authors,

Please provide your response to the following comments/suggestions:

1.     The study exclusively used male Wistar rats for the experiments. It is already known about sex-dependent differences in wound healing and inflammatory responses in both animals and humans, the absence of female subjects limits the generalizability of the findings. A more comprehensive understanding would require the inclusion of both male and female subjects, or an acknowledgment of the potential limitations related to sex bias.

2.     The study's duration appears to be relatively short, as it mentions assessing wound retraction over a limited time frame. Long-term effects of the treatment, including scar formation and tissue quality, should be considered, as they are relevant to the clinical application of wound healing therapies.

3.     When several commercially available products are there for skin wound healing, what is the uniqueness of using the Copaifera langsdorffii? A detailed paragraph is needed for explanation.

4.     While the text introduces important concepts such as nanostructured lipid carriers, it could provide a brief explanation of these terms for readers who may not be familiar with them. Offering a concise definition or a sentence explaining their role in drug delivery would improve comprehension.

5.     Specify the roles of particularly Silvana Tavares Rodrigues. Were they responsible for the plant voucher identification or other aspects of the study? Clarifying their contributions in this study.

6.     There are no animal pictures with the wound. How reviewer can see the changes in the healing process in the experimental groups? Please provide all the group with pictures of wounds and after treatment to show the healing in the animals.

7.     Please provide the pictures for the following statement mentioned in the manuscript “For each sample, the border and the center of the wounds were analyzed and were photographed in five different fields”.

8.     What does the author want to show in Figure 5? There is no description in the figure legend.  Please provide a brief explanation in the figure legend and also mark the area with the arrow that the authors want to show.

9.     Please provide a brief explanation about the figure in Figure 6 and 7 legend. It is very hard to understand what the authors want to show in the figure.

10.  Please provide a schematic diagram of how the wounded or healed area was calculated for better understanding.

11.  What does the greenish color represent in Figures 9 and 10? Need to provide a detailed description. What does it depict in the skin section?

12.  Please provide a detailed description of Figure 12.

13. Is "PBS” is sufficient to homogenize and quantify the protein from the skin samples? Did you add any RIPA, or protease inhibitor or not????

Thank you 

Comments on the Quality of English Language

Please simplify some of the long sentences into smaller sentences. and check the English grammar throughout the manuscript.

Author Response

Dear reviewer,

On behalf of the entire team involved in the production of this manuscript, I would like to thank you for your comments and suggestions that contributed to improving the quality of our manuscript.

Below are the responses to the suggestions of our manuscript.

I remain at your disposal for any questions and clarifications.

Best regards,

Lucas Fernando Sérgio Gushiken

Comments:

Dear authors,

Please provide your response to the following comments/suggestions:

  1. The study exclusively used male Wistar rats for the experiments. It is already known about sex-dependent differences in wound healing and inflammatory responses in both animals and humans, the absence of female subjects limits the generalizability of the findings. A more comprehensive understanding would require the inclusion of both male and female subjects, or an acknowledgment of the potential limitations related to sex bias.

Answer: As reported by the reviewer, the existence of sex-dependent differences in the healing of skin lesions is already known, as well as changes in sexual hormones which result in alteration of wound healing. Furthermore, hormonal variation due to the estrous cycle in rodent females can result in variations in the data of skin wound healing (Gilliver et al., 2007; Horng et al, 2017). Therefore, the use of male rats was made in an attempt to minimize these variations, since the results shown in the article refer to basic research, with the aim of testing a possible healing effect of the formulation, with experiments still to be carried out in the future.

References:

- Gilliver S.C., Ashworth J.J., Ashcroft G.S. The hormonal regulation of cutaneous wound healing. Clin. Dermatol. 2007; 25:56–62. doi:10.1016/j.clindermatol.2006.09.012

- Horng H.C., Chang W.H., Yeh C.C., Huang B.S., Chang C.P., Chen Y.J., Tsui K.H., Wang P.H. Estrogen effects on wound healing. Int. J. Mol. Sci. 2017;18:2325. doi:10.3390/ijms18112325.

  1. The study's duration appears to be relatively short, as it mentions assessing wound retraction over a limited time frame. Long-term effects of the treatment, including scar formation and tissue quality, should be considered, as they are relevant to the clinical application of wound healing therapies.

Answer: The periods of treatment were defined based on the experimental model most used in different articles in recent years (Ali et al., 2023; Kim et al., 2023; Popescu et al., 2023; Sant’Ana et al., 2023; Suh et al., 2023; Zhou et al., 2023). Furthermore, although the model of cutaneous wounds in rodents is widely used because it similarly mimics the healing mechanisms of human injuries and the ease of obtaining samples for biochemical studies, cutaneous injuries in rodents present a mechanism of contraction of the area. of injury more accelerated compared to humans. Therefore, the collection of biological material for analysis after 2 weeks is greatly hampered, with a small area of wound to perform all analysis. Therefore, the periods of 1 and 2 weeks being the most used to carry out analyzes and project the possible effects on healing mechanisms.

References:

- Ali, M.; Kwak, S.H.; Byeon, J.Y.; Choi, H.J. In Vitro and In Vivo Evaluation of Epidermal Growth Factor (EGF) Loaded Alginate-Hyaluronic Acid (AlgHA) Microbeads System for Wound Healing. J Funct Biomater 2023, 14, 403, doi:10.3390/jfb14080403.

- Kim, E.; Seo, S.H.; Hwang, Y.; Ryu, Y.C.; Kim, H.; Lee, K.M.; Lee, J.W.; Park, K.H.; Choi, K.Y. Inhibiting the Cytosolic Function of CXXC5 Accelerates Diabetic Wound Healing by Enhancing Angiogenesis and Skin Repair. Exp Mol Med 2023, doi:10.1038/s12276-023-01064-3.

- Popescu, I.; Constantin, M.; Solcan, G.; Ichim, D.L.; Rata, D.M.; Horodincu, L.; Solcan, C. Composite Hydrogels with Embedded Silver Nanoparticles and Ibuprofen as Wound Dressing. Gels 2023, 9, 654, doi:10.3390/gels9080654.

- Sant´Ana, M.; Amantino, C.F.; Silva, R.A.; Gil, C.D.; Greco, K. V.; Primo, F.L.; Girol, A.P.; Oliani, S.M. Annexin A12–26 Hydrogel Improves Healing Properties in an Experimental Skin Lesion after Induction of Type 1 Diabetes. Biomedicine and Pharmacotherapy 2023, 165, doi:10.1016/j.biopha.2023.115230.

- Suh, J.W.; Lee, K.M.; Ko, E.A.; Yoon, D.S.; Park, K.H.; Kim, H.S.; Yook, J.I.; Kim, N.H.; Lee, J.W. Promoting Angiogenesis and Diabetic Wound Healing through Delivery of Protein Transduction Domain-BMP2 Formulated Nanoparticles with Hydrogel. J Tissue Eng 2023, 14, doi:10.1177/20417314231190641.

- Zhou, C.; Guan, D.; Guo, J.; Niu, S.; Cai, Z.; Li, C.; Qin, C.; Yan, W.; Yang, D. Human Parathyroid Hormone Analog (3-34/29-34) Promotes Wound Re-Epithelialization through Inducing Keratinocyte Migration and Epithelial-Mesenchymal Transition via PTHR1-PI3K/AKT Activation. Cell Commun Signal 2023, 21, 217, doi:10.1186/s12964-023-01243-9.

  1. When several commercially available products are there for skin wound healing, what is the uniqueness of using the Copaifera langsdorffii? A detailed paragraph is needed for explanation.

Answer: In the present work, our group suggests that a new formulation (NLC) presented healing activity in a rat excision wound model, with similar or better results compared to drugs used on the market. Currently available treatments are not able to help in many cases. Therefore, the work is justified in the search for new drugs that have healing activity, acting on different mechanisms of skin wound healing. Furthermore, NLC acted on different mechanisms of the healing process, acting during the different phases of skin wound healing, while currently existing drugs generally act in a single phase.

The information mentioned in the paragraph above was cited in the first paragraph of the “Discussion” section and in the “Conclusions” section.

  1. While the text introduces important concepts such as nanostructured lipid carriers, it could provide a brief explanation of these terms for readers who may not be familiar with them. Offering a concise definition or a sentence explaining their role in drug delivery would improve comprehension.

Answer: The text describing the concepts of nanostructured lipid carriers was inserted in the “Introduction” section and the respective reference was added in the text.

  1. Specify the roles of particularly Silvana Tavares Rodrigues. Were they responsible for the plant voucher identification or other aspects of the study? Clarifying their contributions in this study.

Answer: Silvane Tavares Rodrigues was only responsible for the plant voucher identification. The main text was rewritten to clarify the contributions in the study, as requested by reviewer.

  1. There are no animal pictures with the wound. How reviewer can see the changes in the healing process in the experimental groups? Please provide all the group with pictures of wounds and after treatment to show the healing in the animals.

Answer: The suggested figure was created and inserted as Supplementary material

  1. Please provide the pictures for the following statement mentioned in the manuscript “For each sample, the border and the center of the wounds were analyzed and were photographed in five different fields”.

Answer: A schematic figure of the wounded skin and its respective parts (border and center of the wound and normal skin) was inserted as Supplementary material.

  1. What does the author want to show in Figure 5? There is no description in the figure legend. Please provide a brief explanation in the figure legend and also mark the area with the arrow that the authors want to show.

Answer: The respective figure was altered, highlighting the area of interest and the explanation was inserted in the legend of the figure.

  1. Please provide a brief explanation about the figure in Figure 6 and 7 legend. It is very hard to understand what the authors want to show in the figure

Answer: The explanation was inserted in the respective legends of figures 6 and 7.

  1. Please provide a schematic diagram of how the wounded or healed area was calculated for better understanding.

Answer: The formula of the percentage of wound retraction was widely described in several articles of skin wound healing (Ali et al., 2023; Beserra et al., 2020; Kim et al., 2023; Suh et al., 2023; Zhou et al., 2023). Therefore, we replaced the text and inserted the mathematical formula in order to clarify the understanding.

References:

- Ali, M.; Kwak, S.H.; Byeon, J.Y.; Choi, H.J. In Vitro and In Vivo Evaluation of Epidermal Growth Factor (EGF) Loaded Alginate-Hyaluronic Acid (AlgHA) Microbeads System for Wound Healing. J Funct Biomater 2023, 14, 403, doi:10.3390/jfb14080403.

- Beserra, F.P.B.; Gushiken, L.F.S.; Vieira, A.J.; Bérgamo, D.A.; Bérgamo, P.L.; de Souza, M.O.; Hussni, C.A.; Takahira, R.K.; Nóbrega, R.H.; Martinez, E.R.M.; Jackson, C.J.; Maia, G.L.A.; Rozza, A.L.; Pellizzon, C.H. From Inflammation to Cutaneous Repair: Topical Application of Lupeol Improves Skin Wound Healing in Rats by Modulating the Cytokine Levels, NF-κB, Ki-67, Growth Factor Expression, and Distribution of Collagen Fibers. Int J Mol Sci 2020, doi: 10.3390/ijms21144952

- Kim, E.; Seo, S.H.; Hwang, Y.; Ryu, Y.C.; Kim, H.; Lee, K.M.; Lee, J.W.; Park, K.H.; Choi, K.Y. Inhibiting the Cytosolic Function of CXXC5 Accelerates Diabetic Wound Healing by Enhancing Angiogenesis and Skin Repair. Exp Mol Med 2023, doi:10.1038/s12276-023-01064-3.

- Suh, J.W.; Lee, K.M.; Ko, E.A.; Yoon, D.S.; Park, K.H.; Kim, H.S.; Yook, J.I.; Kim, N.H.; Lee, J.W. Promoting Angiogenesis and Diabetic Wound Healing through Delivery of Protein Transduction Domain-BMP2 Formulated Nanoparticles with Hydrogel. J Tissue Eng 2023, 14, doi:10.1177/20417314231190641.

- Zhou, C.; Guan, D.; Guo, J.; Niu, S.; Cai, Z.; Li, C.; Qin, C.; Yan, W.; Yang, D. Human Parathyroid Hormone Analog (3-34/29-34) Promotes Wound Re-Epithelialization through Inducing Keratinocyte Migration and Epithelial-Mesenchymal Transition via PTHR1-PI3K/AKT Activation. Cell Commun Signal 2023, 21, 217, doi:10.1186/s12964-023-01243-9.

  1. What does the greenish color represent in Figures 9 and 10? Need to provide a detailed description. What does it depict in the skin section?

Answer: An explanation describing what does the greenish color represents was inserted in the section “2.3. Histopathological analysis” and in the legends of “Figure 9” and “Figure 10” to clarify the text, as requested by reviewer.

  1. Please provide a detailed description of Figure 12.

Answer: The legend of “Figure 12” was altered (in the text) to provide a detailed description:

FIGURE 12. Photomicrographs of the immunolabeling of α-SMA in the dermis of wounds treated during 3, 7 and 14 days. Black arrows indicate the immunolabeled myofibroblasts (brown color) positive for α-SMA presented in the center of the dermis. The other cells and extracellular matrix were counterstained with hematoxylin (blue-purple). UT: wounded animal without treatment. NeBa: wounded animal treated with neomycin + sulfate bacitracin zinc. Dex: wounded animals treated with dexpanthenol. Col: wounded animals treated with collagenase. NLC: wounded animals treated with 1% Copaifera langsdorffii oleoresin loaded in nanostructured lipid carriers. Control: animals without lesion and treatment.

  1. Is "PBS” is sufficient to homogenize and quantify the protein from the skin samples? Did you add any RIPA, or protease inhibitor or not????

Answer: A protease inhibitor cocktail was used. The information was inserted in the main text.

Round 2

Reviewer 2 Report

Comments and Suggestions for Authors

Dear Authors,

Thank you for revising the manuscript as per the comments/suggestions. Still, there is a minor modification to include in the manuscript:

Please mention the scale bar for the following figures: 7/8/9/11/12/14/15/16/18/19/20.

Thank you and All the best.

Author Response

Dear Reviewer,

I would like to thank all the suggestions made in our manuscript. Your suggestions certainly helped to improve the quality of the manuscript to be published.
Scale bars were placed on each figure and the indication of their measurement was inserted in the caption of the respective figures.
Thanks again for the comments.

Yours sincerely,

Lucas Gushiken